

# A critical evaluation of decadal solar cycle imprints in the MiKlip historical ensemble simulations

Tobias C. Spiegl[1], Ulrike Langematz[1], Holger Pohlmann[2], Jürgen Kröger[2]

[1]Institute of Meteorology, Freie Universität Berlin, Berlin, Germany

[2]Max Planck Institute for Meteorology, Hamburg, Germany

*Correspondence to*: T. C. Spiegl (tobias.spiegl@met.fu-berlin.de)



**Abstract**
Studies concerning solar-terrestrial connections over the last decades claim to have found evidence that the quasi-
decadal solar cycle can have an influence on the dynamics in the middle atmosphere in the Northern Hemisphere
during winter season. It has been argued that feedbacks between the intensity of the UV part of the solar spectrum and
low latitude stratospheric ozone may produce anomalies in meridional temperature gradients which have the potential
to alter the zonal mean flow in mid to high latitudes. Interactions between the zonal wind and planetary waves can
lead to a downward propagation of the anomalies, produced in the middle atmosphere, down to the troposphere.
More recently it has been proposed that the projection of possible decadal surface solar signals on the North Atlantic
Oscillation might lead to a synchronization of the latter via the 11-year solar cycle. Furthermore, it has been claimed
that a realistic representation of the solar cycle in climate models may lead to a significant increase of the decadal
prediction skill. These conclusions have been debated controversial since then and a confirmation from other
modelling groups is missing.
In this paper we aim for an unbiased evaluation of possible solar imprints from the middle atmosphere to the surface
and with that from head to toe. Thus, we analyze model output from historical ensemble simulations conducted with
the state-of-the-art Earth system model MPI-ESM-HR. The target of these simulations was to isolate the most crucial
model physics to foster basic research on decadal climate prediction and to develop an operational ensemble decadal
prediction system within the MiKlip framework.
Based on correlations and multiple linear regression analysis we show that the MPI-ESM-HR simulates a realistic,
statistically significant and robust shortwave heating rate and temperature response at the tropical stratopause, which
is known from existing studies. However, the dynamical response to this initial radiative signal in the NH during the
boreal winter season is rather weak. In this context we find a slight strengthening of the polar vortex in midwinter
season during solar maximum conditions in the ensemble mean, which basically agrees with the so-called "top-down"
mechanism. The individual ensemble members, however, show a large spread in the dynamical response with opposite
sign in response to the solar cycle.
We also analyze the possible surface responses to the 11-year solar cycle and review the proposed synchronization
between the solar forcing and the North Atlantic Oscillation. We find that the westerly wind anomalies in the lower
troposphere as well as the anomalies in the mean sea level pressure are most likely independent from the seasonal





march in the middle atmosphere since they mimic positive and negative phases of the Arctic- and North Atlantic
Oscillation rather sporadically than in a systematic way.
Finally, by applying lead/lag correlations, our results indicate that the proposed synchronization between the solar
cycle and the decadal component of the North Atlantic Oscillation might rather be interpreted as a statistical artefact
than a plausible physical connection between the solar forcing and quasi-decadal variations in the troposphere.



## 1. **Introduction**

The discipline of decadal climate prediction is rather young and a rapidly growing field in climate science. By using initialized climate model simulations, the gap between weather forecasting and long-term climate model projections covering the complete 21$^{st}$ century or beyond is bridged (e.g. Pohlmann et al., 2013; Meehl et al., 2014). By the aid of decadal climate predictions, policymakers can be equipped with an improved decision-making basis allowing for a better planning of necessary water resources, agriculture, energy and infrastructure measures for the near-term future (Mehta et al., 2011). The aim of the German joint research project "Mittelfristige Klimaprognose" (MiKlip) was to establish a new decadal prediction system allowing for a more precise midterm climate forecasting. To this effect, potential driving factors shaping the decadal climate from both anthropogenic and natural sources have been evaluated critically based on large ensemble simulations with the Max Planck Institute for Meteorology Earth System Model (MPI-ESM).

One factor that potentially influences tropospheric weather and climate is the variability in the middle atmosphere via stratosphere-troposphere coupling processes. The internal variability in the middle atmosphere during the dynamically active winter and spring seasons is strongly controlled by the variability of Rossby waves, which propagate upward from the troposphere to the middle atmosphere where they break and interact with the zonal-mean flow. The changes in the zonal-mean flow, again, can alter the propagation conditions for planetary scale waves initiating a self-consistent feedback called wave-mean flow interaction (e.g. Andrews 1985). As a result, strong disruptions, born in the middle atmosphere, such as sudden stratospheric warmings (SSWs), which are characterized by a breakdown of the polar vortex, have the potential to propagate downward into lower atmospheric layers and interfere with the tropospheric weather regime (e.g. Baldwin and Dunkerton, 2001). A prominent example for this are Northern Hemisphere (NH) cold air outbreaks which have the tendency to be more frequent and severe in seasons with a weak stratospheric polar vortex (e.g. Huang et al., 2021).

A source of variability that might influence the dynamics in the middle atmosphere on the decadal timescale, via a complex feedback between radiation, chemistry and wave-mean flow interaction is thought to be the 11-year solar cycle. Pioneering work concerning the impact of the solar cycle on middle atmosphere dynamics and possible connections to the troposphere goes back to Kodera and Kuroda (2002). Based on a relatively short period of NCEP reanalysis data (1979 – 1998), the authors observed an increase of the tropical stratopause temperature (TST) (at ~50



km) during the time of the solar maximum. In their conceptual explanation, this temperature increase leads to a

strengthening of the meridional temperature gradient and an intensification of the polar night jet (PNJ) in the winter

stratosphere. The stronger westerlies create a barrier for upward propagating planetary waves, which in turn are

deflected poleward and break at lower altitudes. The resulting convergence in the Eliassen-Palm flux (EPF) allows the

positive wind anomaly to penetrate downward, where it is dragged towards the pole over the winter season. Kodera

(2002) argues that the solar induced wind anomalies may advance into the troposphere, where they create a signal in

meteorological variables mimicking a positive phase of the North Atlantic Oscillation (NAO). Matthes et al. (2004,

2006) studied the proposed "top-down" mechanism by the aid of idealized simulations with an early middle

atmosphere 3-dimensional general circulation model (GCM). They found that during solar maximum conditions the

polar vortex seems to be stronger especially in November and December and linked this to a positive Arctic oscillation

(AO)-like pattern which they found in lower altitudes and to some extent at the surface. The observed pattern weakens

in January and changes sign from February on. In subsequent studies comparable results have been found (e.g., Marsh

et al., 2007; Schmidt et al., 2010; Ineson et al., 2011; Chiodo et al., 2012; Langematz et al., 2013), however, showing

very individual temporal progressions of the signals from the middle atmosphere to the surface varying from December

to February and the proposed influence in the North Atlantic region. These earlier studies are often quoted as

convincing proof for a "top-down" influence of the 11-year solar cycle in both the middle atmosphere and the

troposphere. Complementary to this, Gray et al. (2013) found that the strongest NAO-like solar-induced signals in the

North Atlantic (i.e. a positive phase of the NAO) actually seem to appear with a time lag of three to four years after

the solar maximum in the respective seasonal winter mean (DJF). However, the observed lags could not be reproduced

in coupled atmosphere-ocean simulations conducted by the same group. In the model, the postulated response to the

solar cycle in the North Atlantic appears almost in phase with the solar forcing (maximum response between lag year

zero to one)  (Gray et al., 2013). This discrepancy between observed and simulated lag in the response in the North

Atlantic NAO was confirmed in subsequent studies (e.g. Scaife et al., 2013; Andrews et al., 2015).

With respect to possible solar induced impacts on NH surface variability in the winter season, Thiéblemont et al.

(2015) went one step further. Analyzing a simulation incorporating 150 model years, they claim that the solar forcing

synchronizes the decadal component of the NAO variability spectrum, a phase relation they can not find in an

experiment without 11-year solar variability. This result has been debated controversially since its publication. Chiodo

et al. (2019) found almost identical spectra of the NAO decadal variability in two simulations of 500 model years each,





with and without a 11-year solar cycle forcing. Furthermore, they identified NAO patterns in similar time segments in
both experiments (forced and unforced). They suspect, therefore, that the alleged surface solar signals in other studies
are most likely a result of the internal variability of the NAO itself rather than solar cycle imprints. On the other hand,
Drews et al. (2022) most recently argue that the solar cycle near-surface imprints can only shine through during very
active solar periods with large amplitudes of the 11-year solar cycle. They also state that during these periods the
surface decadal prediction skill would be significantly enhanced if the solar cycle is a vital part of the prediction
system.
In this publication, we evaluate possible imprints of the 11-year solar cycle in different domains of the atmosphere
from the initial solar radiative signal in the tropical upper stratosphere down to the surface in the NH winter season.
We analyze the MiKlip historical ensemble simulations conducted with the state-of-the-art Earth system model MPI-
ESM-HR, which is the physical basis for the decadal prediction system, which is operational at the "Deutscher
Wetterdienst" (DWD) since 2020. The availability of the large amount of output data from the MiKlip historical model
ensemble enables us to address the unresolved questions of the solar surface imprint on a more robust statistical basis
than is possible in single model simulations. In our study, we aim to identify the role of the solar imprints for the
decadal variability of the NAO in winter.
This publication is structured as follows. In Section 2 we describe the MPI-ESM, the setup of the analyzed simulations
and the applied methodologies to detect potential solar cycle signals in different atmospheric domains. In Section 3,
the initial radiative solar signal in the tropical middle atmosphere is evaluated. Subsequently, we concentrate on the
dynamical response to the initial solar signal in the NH winter season. Here we show in Section 4 the ensemble mean
response and compare individual ensemble members with opposite solar signatures. In Section 5, we derive solar-
induced signals near the surface in our simulations and observations. In Section 6, we check our model results with
respect to the proposed synchronization between the solar forcing and the NAO. Finally, we summarize and discuss
our results in a broader context (Section 7).



## 2.  **Data and methods**

2.1 Model description and experimental design

The historical simulations analyzed in this publication  have been conducted with the Max Planck Institute for Meteorology Earth System Model in high resolution configuration (MPI-ESM1.2-HR; hereafter called MPI-ESM-HR) at the Deutsches Klimarechenzentrum (DKRZ). MPI-ESM-HR includes the atmospheric general circulation model ECHAM (European Centre Hamburg) version 6.3 (ECHAM6.3) with a horizontal/vertical resolution of T127L95 (corresponds to a ~100 km * 100 km model grid and 95 levels in the vertical with a model top at 0.01 hPa or ~80 km) (Müller et al., 2018). The high vertical resolution allows for an internally generated quasi-biennial oscillation (QBO) in the tropical stratosphere (Pohlmann et al., 2019). Radiative processes are represented using the rapid radiation transfer model for GCMs (RRTM-G) for both the shortwave and longwave part of the electromagnetic spectrum (Iacono et al., 2008). Other diabatic processes, such as vertical mixing by turbulence and moist convection, large-scale convection, and momentum deposition by orographic and unresolved gravity waves are described in more detail in Stevens et al. (2013). Oceanic processes are accounted for in the coupled Max Planck Institute ocean model (MPIOM) with a TP0.4 (0.4° nominal) resolution (Jungclaus et al., 2013). MPI-ESM-HR further incorporates the biogeochemistry module Hamburg Model of the Ocean Carbon Cycle (HAMOCC) (Ilyina et al., 2013; Paulsen et al., 2017) and the land surface model JSBACH (Reick et al., 2013).

In this publication, we analyze 10 members of the MPI-ESM-HR historical simulations performed within the German research project MiKlip. The MiKlip historical ensemble simulations include the observed natural and anthropogenic climate drivers, as described in the CMIP5 protocol (Taylor et al., 2013). The individual ensemble members (1 to 10) have been initialized from different model years of a 1850 preindustrial (PI) control simulation and were integrated over the period 1850 to 2005. Here, we focus on the period 1880 – 1999. Thus, a total of 1,200 model years have been evaluated. Since the model does not include interactive atmospheric chemistry, ozone concentrations have to be prescribed. In the MiKlip historical simulations, the merged CMIP5 ozone dataset was used, which consists of a combination of SAGE I+II satellite and radiosonde data in the period 1979 to 2005. To derive earlier ozone concentrations back to 1850, the zonal mean stratospheric time series is extended backwards based on the regression



fits and proxy time series of equivalent effective stratospheric chlorine (EESC) and solar variability (Cionni et al.,
2011). The solar variability forcing includes all observed solar cycles and follows Lean (2000).

2.1 Data analysis
*Detrending, correlations, filtering*
To detrend the sunspot number (SSN) (Source: WDC-SILSO, Royal Observatory of Belgium, Brussels -
https://www.sidc.be/silso/infosnmtot)  and shortwave heating rate time series, a third-degree polynomial function has
been fitted to the data,  the respective anomalies are shown in Figure 1. The detrended SSN time series has then been
correlated (Pearson r) with the detrended tropical stratopause temperature (defined as the mean value between 25°S
– 25°N at 1 hPa (Figure 3)). To reduce the degree of internal variability, a Butterworth bandpass filter with cutoff
frequencies of 9 and 13 years has been applied to the detrended PNJ time series (defined as the arithmetic mean of
the zonal-mean zonal wind between 35°N – 45°N at 1 hPa) (Figure 3). The same Butterworth bandpass filter has
also been applied to the zonal-mean zonal wind time series at 10 hPa (zonal mean over 55°N – 65°N) (Figure 3) and
the NAO time series. The NAO time series has been calculated by the aid of an EOF analysis conducted for the
MSLP data over the Atlantic sector (20 – 80°N, 90°W – 40°E) in the winter season (DJF averaged and individually
for December, January and February). The first principal component is then used to describe the NAO variability.
The lead/lag correlations (Figure 8) are then calculated between the filtered NAO and SSN time series.
*Multiple linear regression*
To detect the solar cycle signals in the middle atmosphere (Figures 2, 4 and 5) and in the mean sea level pressure in
both observations and model data (Figures 6 and 7), we use an established multiple linear regression (MLR) technique
as described in Bodeker et al. (1998). To derive the individual regression coefficients, we use a set of six predictors in
the MLR model:
$X(t) = Off.const + A * CO_2(t) + B * QBO(t) + C * QBOorth(t) + D * SSN(t) + E * Nino3.4(t) + F * tau(t) + R(t)$
with:



*Off.const = annual cycle; CO2(t) = increase in the atmospheric $CO_2$ concentrations, QBO(t) = phase of the QBO*
*defined via the zonal-mean zonal wind in 30 hPa (5°S – 5°N); QBOorth(t) = the orthogonal of QBO(t); SSN(t) = SSN*
*time series; Nino3.4(t) = Nino3.4 times eries; tau(t) = optical thickness at 550 nm and R(t) = model residuum*
The solar related coefficients have been scaled to 180 SSN, which is a good approximation for a mean solar cycle
amplitude. To detect potential time lags in the response to the solar cycle at the surface, the solar time series has been
shifted in such a way that the solar forcing lags the model data between one and four years.

## 3.   The initial radiative solar signal in MPI-ESM

The dynamical "top-down" mechanism, assumed to be the pathway for the propagation of the solar signature through
the atmosphere to the surface in NH winter (see also Section 1), is initiated at the tropical upper stratosphere by the
absorption of solar ultraviolet (UV) irradiance by ozone and molecular oxygen. In particular, the absorption of solar
photons by ozone in the Hartley bands (200 – 310 nm) in the upper stratosphere - and to a lesser extent the Huggins-
bands (310 nm – 400 nm) in the middle stratosphere – heats the upper stratosphere increasingly with height and leads
to the formation of the warm stratopause. Although the variation in solar UV-irradiance over the 11-year solar cycle
is less than 10% in the ozone absorption bands, the enhanced UV radiation at solar maximum – in combination with
increased ozone concentrations - leads to stronger shortwave heating and a concurrent warming of the tropical
stratopause by the order of 1 K, as has been derived from merged MSU4 and SSU+MLS-satellite observations (Randel
et al., 2016).
Figure 1a shows the annual mean response of the modelled shortwave radiative heating rate (SWHR) at the
stratosphere and lower mesosphere (100 – 0.1 hPa) for a range of solar cycle (SC) amplitudes from the weak SC14 (in
blue), over the medium SC22 which has been used as solar forcing in the CMIP5 protocol (in green), to the very strong
SC19 (in red). MPI-EMS-HR produces the well-known solar cycle impact with enhanced SW heating during solar
maximum throughout the upper stratosphere and lower mesosphere. The maximum SWHR difference develops at the
stratopause and ranges for the three selected solar cycles between 0.17 and 0.51 K/day. With a SWHR increase of 0.32
K/day for the SC22 solar forcing, MPI-ESM-HR produces an initial solar radiative response at the tropical stratopause
which is in very good agreement with offline radiation model calculations using the CMIP5 solar forcing (i.e. the same




forcing as in MPI-ESM-HR) in a line-by-line reference and two CCM (EMAC and WACCM) radiation codes (see
Figure 8, yellow curves in Matthes et al., 2017). This is a significant improvement compared to the earlier ECHAM4
and ECHAM5 model versions which were not able to simulate the SWHR response to the solar cycle in the
stratosphere (see Figure 17 in Forster et al., 2011), and thus missed the initial solar temperature signal necessary for
the "top-down" mechanism. The improvement in the MPI-ESM-HR is the result of the enhanced spectral resolution
of the new shortwave radiation scheme in ECHAM6 which resolves the shortwave spectrum in 14 bands spanning the
wavelength range from 820 to 50,000 cm$^{-1}$ (Iacono et al., 2008), whereas ECHAM4 and ECHAM5 used a lower
spectral resolution with the four-band model of Fouquart and Bonnel (1980), later extended to six bands by Cagnazzo
et al. (2007).
Figure 1b shows the time series of the SSN and the modeled SWHR at the tropical stratopause over the full simulated
period from 1850 – 1999. The shown anomalies of both time series from a third-degree polynomial fit clearly
demonstrate that solar cycles of different amplitudes initiate SWHR responses that closely follow in magnitude the
strength of the solar forcing. The only exception is found for the weak SC20 which starts from a pretty high minimum.
This is not reproduced in the SWHR, possibly due to the transition from synthetic SSN before 1979 to observed SSN
afterwards.

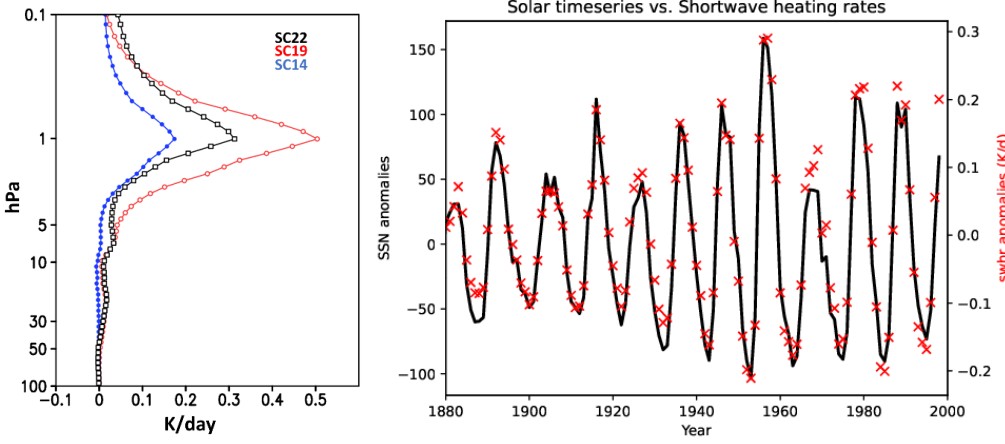


**Figure 1:** Solar shortwave heating rate signature in the MPI-ESM-HR historical simulations: a) Annual tropical
mean (25°S – 25°N) shortwave heating rate difference in K/day between the maximum and minimum of three solar
cycles: the weak solar cycle 14 (blue), the medium solar cycle 22 used in CMIP5 (green), and the strong solar cycle
19 (red) (a), and: Time series of the sunspot number and the annual tropical mean (25°S – 25°N) shortwave heating
rate at the stratopause (1 hPa). Shown are anomalies from a third-degree polynomial fit to the data (b).






When averaging over all solar cycles between 1880 and 1999 and all 10 ensemble members, we receive a robust,
highly significant annual mean warming of the complete middle atmosphere at solar maximum (Figure 2a), reaching
a peak response of 1.2 K at the tropical stratopause (Figure 2b). This result is slightly higher than the solar signal
derived from satellite observations (0.7 K/ (100 solar flux units)-1, respectively ~1 K between solar minimum and
maximum) (Randel et al., 2016), which is probably due to the relative short time series of satellite observations
compared to the simulated time series.
Given the excellent temporal evolution of the initial radiative response of the upper tropical stratosphere to the decadal
solar forcing, we conclude that MPI-ESM-HR produces the necessary prerequisite for the dynamically enhanced "top-
down"-mechanism, which will be investigated in more detail in the next section.

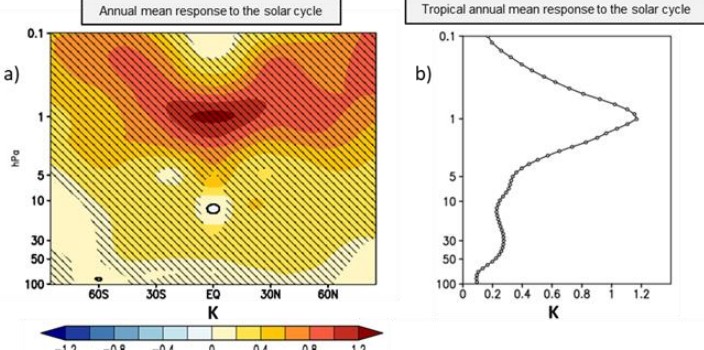

**Figure 2:** Long-term annual ensemble mean response of the zonal-mean temperature (in K) to the solar cycle in the
middle atmosphere as a function of height and latitude (hatched regions mark the 95% level of significance) (a), and
the annual mean tropical (25°S – 25°N) temperature response (in K).


**4. Downward transfer of the solar signal to the surface: the key role of dynamics**

After having demonstrated the ability of the MPI-ESM-HR model to realistically simulate the radiative and the related
temperature response in the tropical upper stratosphere to the decadal solar forcing, we investigate as next step the
potential dynamical reaction to the radiative forcing, which is expected according to the "top-down" mechanism. By
evaluating the ensemble spread in the NH during the dynamically active season (November to March), we assess the





variability of different dynamical variables in the stratosphere with respect to the solar fluctuations in the MPI-ESM-
HR historical ensemble simulations. We focus first on the detrended deviations from the long-term monthly means for
the TST and (to estimate the dynamical response in the NH) the zonal-mean zonal wind at two different altitudes and
latitudes (Figure 3). To approximate the PNJ (the local maximum wind speed in the upper stratosphere) we use the
mean of the zonal-mean zonal wind in 35°– 45°N at 1 hPa. The variability in the middle stratosphere is represented
by the mean of the zonal-mean zonal wind in 55°– 65°N at 10 hPa. After calculating the respective anomaly time
series for the TST, the PNJ and the 10 hPa zonal wind variations for each month individually, we correlate these time
series with the detrended DJF mean SSN time series. To mute the interannual variability (operating on timescales
between 1 and 8 years) of the polar vortex, the PNJ and 10 hPa anomaly time series, as well as the SSN time series,
have been bandpass-filtered, before calculating the correlations. Our results indicate that the TST correlates
significantly with the SSN, not only in the annual mean (compare  Figure 1b) but  also in each individual month
considered (Figure 3, left column). While negative and positive TST anomalies  (i.e. negative and positive deviations
from the long-term monthly mean) are almost uniformly distributed for SSN values smaller than  the SC14 maximum
(blue dotted lines), an increase in the solar forcing exceeding the SC14 SSN maximum leads to a higher probability of
positive TST anomalies. The strength of the correlations changes over the season, such that a stronger connection
between the solar forcing and the temperature response at the tropical stratopause is given in late autumn (November:
r=0.28) and late winter (February: r=0.34; March: r=0.42). In these months, a particular strong solar forcing (indicated
by the SSN value of the SC19 maximum (red dotted lines)) is almost always associated with a positive temperature
anomaly at the tropical stratopause. Weaker correlations and a broader distribution of negative and positive
temperature anomalies, even during periods with especially pronounced solar activity, are calculated for the midwinter
season (December: r=0.15; January: r=0.16). These findings are consistent with an increase in the overall variability
in the TST during December and January, making it more difficult for the relatively weak solar induced signals to be
distinguished from the background noise. The higher variability in the TST during December and January is probably
a result of the higher variability of the tropical branch of the Brewer-Dobson circulation (BDC) in boreal winter.
According to the general concept of the "top-down" mechanism the initial signal in the TST would be accompanied
by a strengthening of the PNJ via a modification of the meridional temperature gradients. Considering the statistically
significant temperature signals and correlations at the tropical stratopause in the MPI-ESM-HR model (Figure 3, left
column), we expect a dynamical response of the PNJ in our simulations. However, the correlations between the SSN



and the PNJ time series (Figure 3, middle column) do not show statistically meaningful relations between the solar
forcing and the dynamical response of the PNJ. Only during February, a weak but statistically significant correlation
is found, which might be related to the enhanced impact of the solar forcing in the TST during the same month.
However, this connection as well becomes insignificant, if the correlations are calculated based on the unfiltered SSN
and PNJ time series. Figure 3 (right column) shows the correlations between the solar forcing and the zonal mean
zonal wind for the lower (and more northward) 10 hPa anomaly time series. We find the strongest (and significant)
correlations in November (r=0.25)  and December (r=0.13), although these correlations become (again) negligible if
the correlations are calculated based on unfiltered model data. The differences in the timing between the maximum
correlations of the SSN with the PNJ (February) and the 10 hPa zonal wind time series (November and December) are
not in line with the established idea of a successive "poleward and downward" progression of the dynamical solar
signal. Furthermore, the computed SSN/PNJ correlations for November, December, January and March are $\leq 0.06$,
implying that the characteristics of the PNJ are not markedly influenced by the magnitude of the solar forcing and thus
the amplitude of the solar cycle.



**Figure 3:** Scatter diagram of the stratopause temperature (left column), PNJ (middle column) and zonal-mean zonal wind averaged over 55°N – 65°N at 10 hPa (right column) variations vs. SSN. The numbers given in the headings show the correlation coefficients (r), their statistical significance (p < 0.05: significant correlation, or p > 0.05: insignificant correlation), and the overall variation (σ). The dotted blue and red lines indicate the SSN at solar cycle maximum for SC14 and SC19 (the weakest/strongest solar cycles considered in the simulations).

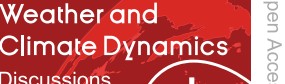 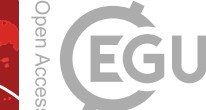

Figure 3 demonstrates that while the connection between the solar forcing and the TST is clearly visible in our
correlation analysis, the potential dynamical response in the NH is harder to detect, especially due to the highly variable
polar vortex. Therefore, we proceed using a MLR analysis to separate the potential dynamical solar induced signals
from other internal generated disturbances in the ensemble mean.
Figure 4 shows the solar regression coefficients, scaled to a mean amplitude of the solar cycle (180 SSN), for the
zonal-mean temperature (top row), the zonal-mean zonal wind (middle row) and the EPF (vectors) and its divergence
EPFD (colors) (bottom row) for each NH winter month (November – March). Here, we focus on the potential solar
cycle signals between the equator and the North Pole and pressure heights in 1.000 hPa – 0.1 hPa for the temperature
and wind responses and 100 hPa – 0.1 hPa for the EPF diagnostics. We find a significant response in the zonal mean
temperature at the tropical stratopause (Figure 4, top row) with a maximum response at the equator of 1.2 K during
November. The solar induced temperature signal is confined to the inner tropics in late autumn and early winter and
advances towards higher latitudes between January and March. This is consistent with the seasonal march of the
incidence angle of solar radiation after the winter solstice in December. In the middle to polar latitudes we find a clear
dipole in the temperature anomalies especially during November and December. This dipole is characterized by
distinct (and significant) positive temperature anomalies in the lower mesosphere and upper stratosphere and weak
(and insignificant) negative anomalies in the middle and lower stratosphere. Particularly the pronounced polar heating
in the upper stratosphere from November to December agrees well with a most recent analysis of ERA-interim
reanalysis data by Kuroda et al., (2022). The detected temperature signals in the middle atmosphere in November and
December are in line with the anomalies in the zonal-mean zonal wind (Figure 4, middle row), which indicate a
stronger (and thus cooler) polar vortex during these months. Additionally, a convergence of the EPF (indicated by the
reddish colors in Figure 4, bottom row) and its (here downward oriented) vectors imply a reduced upward propagation
of planetary waves due to the strengthening of the polar vortex. The maximum (and significant) response in the
stratospheric zonal-mean zonal wind in the area of the polar vortex, is located at ~60°N at 10 hPa. Here, we find
positive anomalies of the zonal-mean zonal wind of ~1 m/s. Given the mean zonal-wind speeds between 20 m/s
(November) and 30 m/s (December), simulated by the model (not shown) at this height and latitude, the solar influence
seems rather small in comparison. The detected dipole in the zonal-mean temperature starts to weaken from January
on and vanishes almost completely until March. During the same months, we find a (yet insignificant) weakening of
the polar vortex which allows for more upward propagation of planetary waves (indicated by a divergence of the EPF



(bluish colors) and upward oriented vectors). In the troposphere, a weak (≤ 0.5 m/s) but significant westerly wind
anomaly around ~60°N can be detected in November and December. The weak tropospheric wind response agrees
with other studies (Matthes et al., 2006; Schmidt et al., 2010; Ineson et al., 2011; Chiodo et al., 2012; Langematz et
al., 2013; Kuroda et al., 2022; Drews et al., 2022).
While in some studies the march of the westerly wind anomalies from the middle atmosphere to the surface seems to
follow the proposed "poleward and downward" concept (e.g. Matthes et al., 2006; Ineson et al., 2011; Drews et al.,
2022), the signal transmission in the MPI-ESM-HR and other model simulations (e.g. Schmidt et al., 2010; Chiodo et
al., 2012; Kuroda et al., 2022) rather follows a "downward-only" storyline. Additionally, the description of the
westerly wind anomalies at the surface is sometimes inconsistent with the idea of a successive downward propagation
of the signal from higher to lower altitudes. As an example, significant westerly wind anomalies at the surface at
middle latitudes are already present in November in the modeling studies of Matthes et al. (2006) and Kuroda et al.
(2022), even though the major signal is still high up in the middle atmosphere. Furthermore, in Kuroda et al. (2022)
the westerly wind anomalies at the surface at middle latitudes are present throughout the complete season (i.e. in all
months between November-March), similar to our MPI-ESM-HR simulations. In other studies, the westerly anomalies
are insignificant (e.g. Schmidt et al., 2010) or do not reach the ground (e.g. Chiodo et al., 2012). This implies that the
detected surface wind anomalies could be independent from the seasonal march in the middle atmosphere and might
rather be a product of the internal variability in the troposphere (i.e. the AO or NAO) itself. Likewise, the temperature
response to the solar cycle in the troposphere with positive temperature anomalies of ≤ 0.2 K at the surface is rather
weak (Figure 4, top row). Interestingly, these small temperature signals are significant in the tropics in all considered
months, which is consistent with the high (and relatively constant) solar insulation in the inner tropics and a damped
overall variability compared to the extratropical regions. By contrast, the significant surface temperature anomalies in
the extratropical regions are located between 50°N and 60°N until January and shift towards the polar latitudes in
February and March.




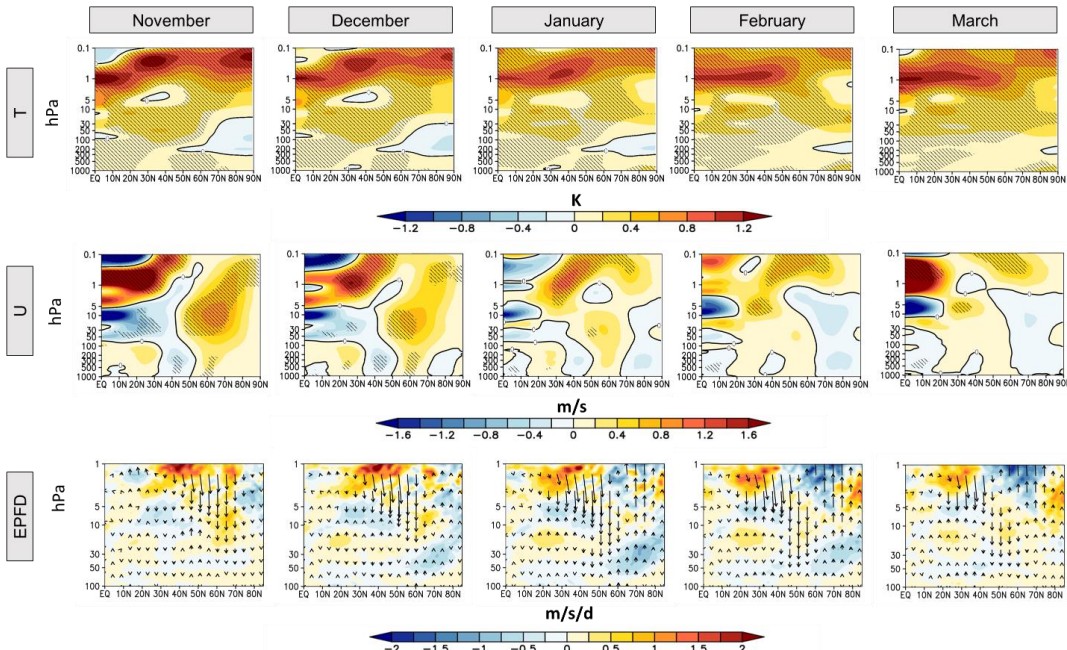

**Figure 4:** The ensemble mean long-term response to the solar cycle of the zonal-mean temperature (first row), zonal-mean zonal wind (second row) (hatched regions mark the 95% level of significance), and the EPF (vectors) and the divergence of the EPF (EPFD, colors) in the NH during the boreal winter season.

So far, we focussed on the discussion of the potential solar signals in the ensemble mean derived from the 10 individual

MiKlip historical simulations thus obtaining statistically more robust results than is possible through analyses of single

simulations. The necessity of working with ensemble mean results is impressively demonstrated by comparing two of

our 10 individual ensemble members. Figure 5 shows the solar regression coefficients for the zonal-mean temperature

and zonal-mean zonal wind for the ensemble members 1 (EM1, top panel) and 4 (EM4, bottom panel), as in Figure 4.

The derived patterns for the solar zonal-mean temperature signal in EM1 show distinct similarities with the ensemble

mean. As an example, we find a (significant) maximum temperature response around the tropical stratopause.

Furthermore, the distribution of the temperature anomalies in the middle to higher latitudes again displays the polar

heating in the lower mesosphere and the upper stratosphere and the cooling in the middle to lower stratosphere. Again,

this pattern starts to weaken from January on. We notice that in comparison to the ensemble mean, fewer areas depict

significant temperature signals, even though the magnitude of the temperature response is stronger. This can be



attributed to the fact that the analysis only includes 120 model years and thus ~12 solar cycles (instead of 1.200 and
~120 in the ensemble mean), which is seemingly not enough to dampen the internal variability and inhibits the solar
induced signals to become significant against the overall background noise. Likewise, the solar response of the zonal-
mean zonal wind in the middle atmosphere in EM1 shows the main characteristics, as already noticed in the ensemble
mean, such as a strengthening of the polar vortex in November and December and a subsequent weakening and a
conversion in sign afterwards. However, none of the detected signals in the area of the polar vortex are statistically
significant. As for the response of the zonal-mean zonal wind at the surface, we detect significant anomalies in January
and February. The geographical distribution of the anomalies (westerly wind anomalies at middle latitudes and easterly
wind anomalies at polar latitudes), however, mimic a negative phase of the AO which is not in line with the general
concept of solar induced "top-down" influences.
In EM4, the initial temperature signal in the upper tropical stratosphere is, as in EM1, visible throughout the complete
season and the strongest in November and December. Thus, the response to the solar cycle in these latitudes and
heights turns out to be a robust feature in The MPI-ESM-HR model experiments. However, even though exactly the
same solar forcing has been applied in EM4 as in EM1, the dynamical response of EM4 looks very different. For
instance, we find a cooling of the polar upper stratosphere and a (significant) warming in the middle to lower
stratosphere in December and January. This pattern is  common during SSWs, which (by chance) could have been
more frequent in EM4 during December and January than in EM1. The strong and significant easterly wind anomalies
in the middle atmosphere, indicating a slowdown of the polar vortex during these months, underpin this hypothesis.
These findings imply that the detected signals in EM1 could also be a result of (by chance) less frequent SSWs in EM1
leading to a potentially misleading attribution to solar variability. In our simulations, four out of 10 simulations show
a weakening of the polar vortex during high solar activity, while six depict a strengthening of the latter, which may
explain the rather weak tendency to westerly wind anomalies in the ensemble mean.
Either way, our results point to the fact that the internal dynamics of the polar vortex have the ability to control the
transmission of potential solar induced signals from the tropics to the polar regions and are thus more important than
the amplitudes of individual solar cycles (compare also Figure 3), as recently claimed by Drews et al., (2022).

## Ensemble member 1

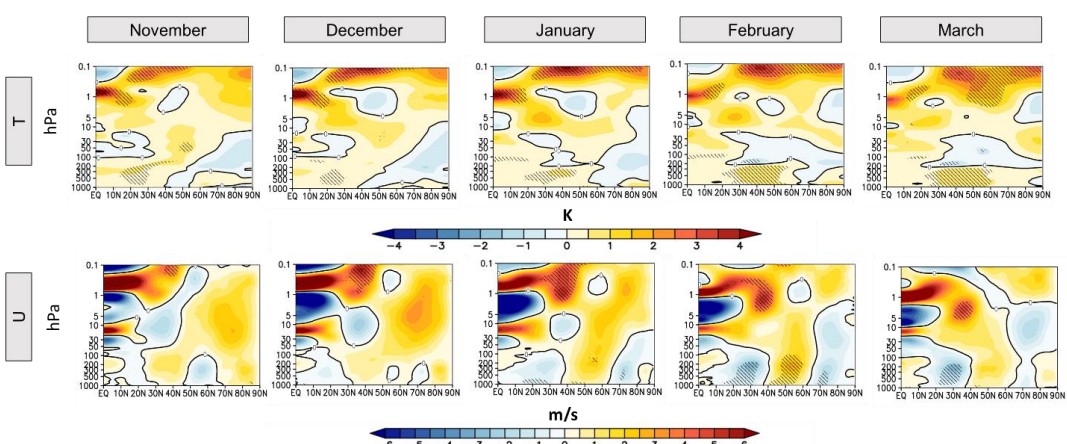

## Ensemble member 4

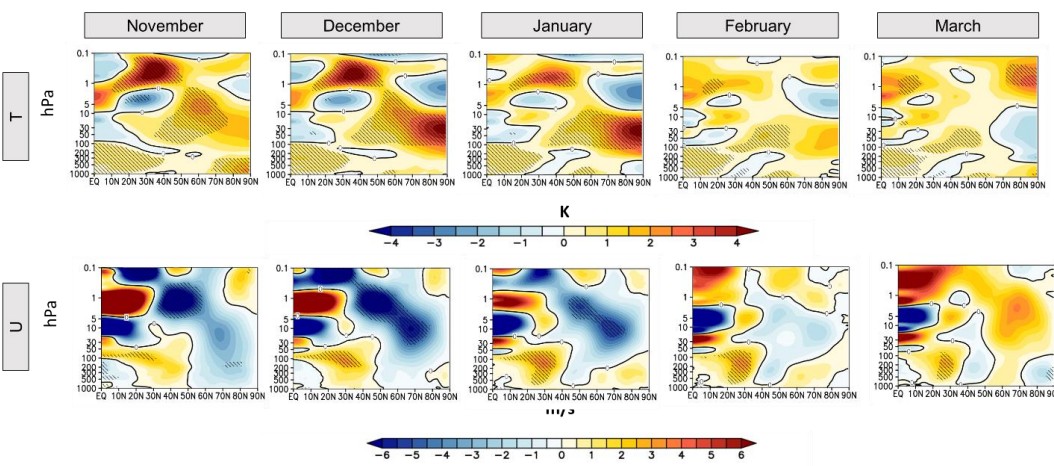


**Figure 5:** Long-term response to the solar cycle of the zonal-mean temperature (first row) and the zonal-mean zonal wind (second row) (hatched regions mark the 95% level of significance) in the two ensemble members EM1 (top panels) and EM4 (bottom panels) in the NH during the boreal winter season.










**5. Direct and lagged surface solar signals**

Our results so far indicate a robust response of the TST to the quasi-decadal solar cycle. The subsequent dynamical response in the NH during the boreal winter season, however, is difficult to assess. By the aid of a MLR analysis we could detect weak solar cycle imprints in the zonal-mean temperature and the zonal-mean zonal wind in the ensemble mean. However, these signals are not robust among all individual ensemble members, especially with respect to the detected anomalies in the zonal-mean zonal wind at the surface which seem to be independent of the signals in the middle atmosphere.

Nevertheless, in the next step, we first aim at detecting potential solar signals at the surface by applying the MLR analysis to mean sea level pressure (MSLP) data in NH winter. Figure 6 shows the monthly solar regression coefficients for MSLP, scaled to a mean solar cycle amplitude of 180 SSN, in the HadSLP2 observational dataset (Allan and Ansell, 2006) for the same period as simulated (1880 – 1999). In order to check for eventual time lags between the applied solar forcing and the model response, as suggested for example by Gray et al. (2013), lagged regressions were calculated by shifting the solar predictor time series against the observations so that it leads the model data between one and four years. Our results show positive and negative anomalies in the MSLP in the middle and polar latitudes which mimic positive and negative phases of the AO in a rather random than systematic way. As an example, we find an AO-positive like pattern (i.e., negative pressure anomalies over the North Pole and positive pressure anomalies in the surrounding middle latitudes) in November at lag year four, in December at lag year four, in February at the lag years one to three and in March at lag year one. The most pronounced AO-positive anomalies, with a negative but insignificant anomaly of ~2 hPa over the North Pole and a positive anomaly of the same magnitude in the middle latitudes, are given at lag year 2. Hence, the strength of the detected potential solar signals in our HadSLP2 analysis is in line with other studies assessing observational products (e.g., Gray et al., 2013; Kuroda et al., 2022; Drews et al., 2022). The detected maximum impact at lag year 2 in February in our analysis, however, agrees with Kuroda et al. (2022) and Drews et al. (2022) but differs from Gray et al. (2013) who found a maximum response at lag year 4 in the DJF mean. These discrepancies in the timing of the peak solar-induced surface signal in the HadSLP MSLP data can only be explained by differences in the analysis techniques, and reveal a high sensitivity of solar-induced surface signals to the applied methodology and individual interpretation of the results.

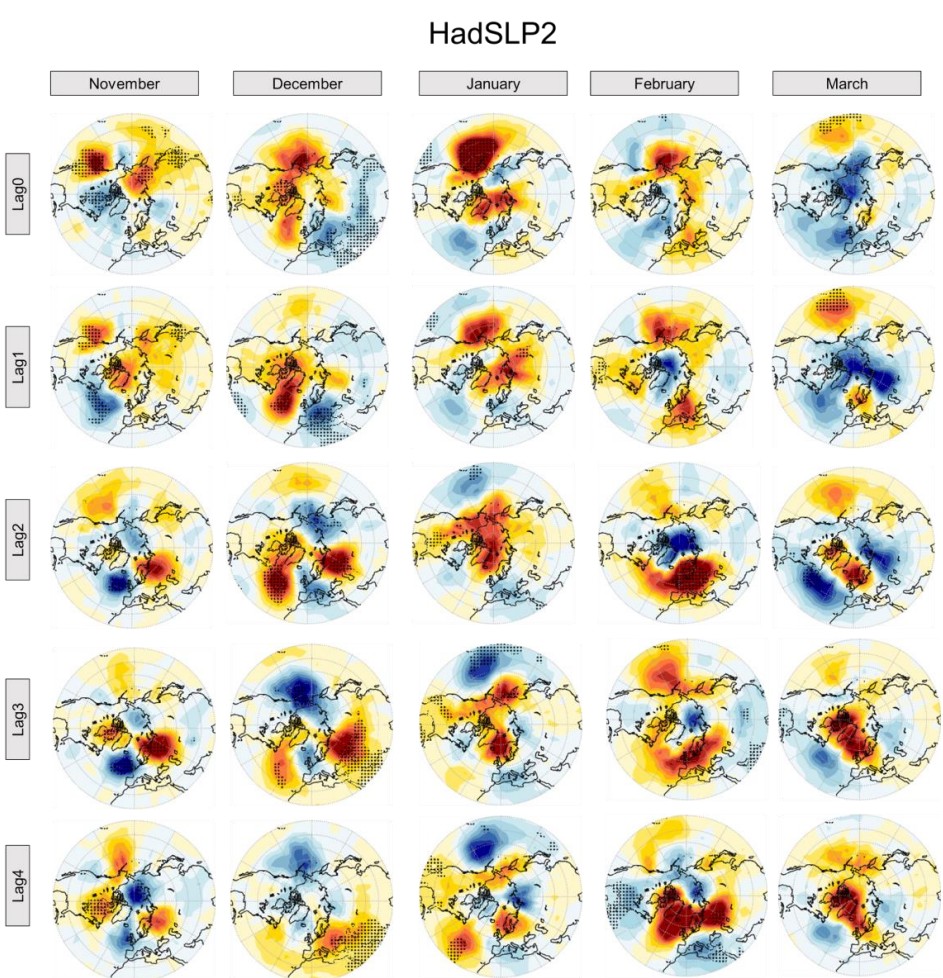

420

**Figure 6:** The (lagged) response of mean sea level pressure (MSLP) to the solar cycle in the NH during the boreal winter season for the HadSLP2 dataset (dotted regions mark the 95% level of significance). Columns denote the individual months of the winter season; rows indicate the lag of the MSLP time series with respect to the solar forcing time series.



Figure 7 shows the same analysis for the MiKlip historical simulations, i.e. the ensemble mean of the solar regression
coefficients for the MSLP for each month (November to March) and the (lag) years zero to four. We detect AO-
positive-like  anomalies in the MSLP in December at the lag years 0 and 1, in January at the lag years 0 to 4 and in
February at the lag years zero to four. The strongest negative MSLP anomalies over the North Pole show a response
of ~ -1.5 hPa and ~ +1.5 hPa in the middle latitudes in January and December. Thus, the overall model response is
weaker compared to the observational data. This is not surprising given the fact that the model results depict the mean
over 10 ensemble members (with respective dampening effects) compared to one 'ensemble member' representing the
observations. While the detected magnitudes of the MSLP anomalies in MPI-ESM-HR agree with other solar cycle
model studies (e.g. Gray et., 2013; Scaife et al., 2013; Andrews et al., 2015; Drews et al., 2022), the detected timing
(i.e. the progression of the signals from the middle atmosphere to the surface) in the MPI-ESM-HR does not fit the
narrative of the "top-down" mechanism as described most recently by Kuroda et al. (2022) and Drews et al. (2022).
In these studies, the authors find the most pronounced AO-positive like pattern in February at the surface and link this
to the coupling between the stratosphere and the troposphere, which peaks in exactly this month. In contrast, in our
model simulations the strongest coupling between the stratosphere and the troposphere appears in December (see
Figure 4), while the most pronounced AO-positive like patterns appear in January and February at different lag years.
We, therefore, assume that the detected surface solar signals could rather be a product of the internal variability in the
troposphere itself than being necessarily a consequence of the proposed "top-down" mechanism. Even if we assume
that the detected surface signals have a pure solar source (and the "top-down" mechanism is always present during
solar maximum years) it seems to be questionable in our view if these tiny signals would have the capability to
synchronize powerful large-scale climate modes such as the AO or the NAO, if they only emerge once per decade
over the duration of a month. As an example, the Icelandic Low and the Azores High, both controlling the pressure
gradients in the North Atlantic sector, show a month by month variation of  ~8.5 hPa and ~6 hPa during winter time
in the model (not shown).

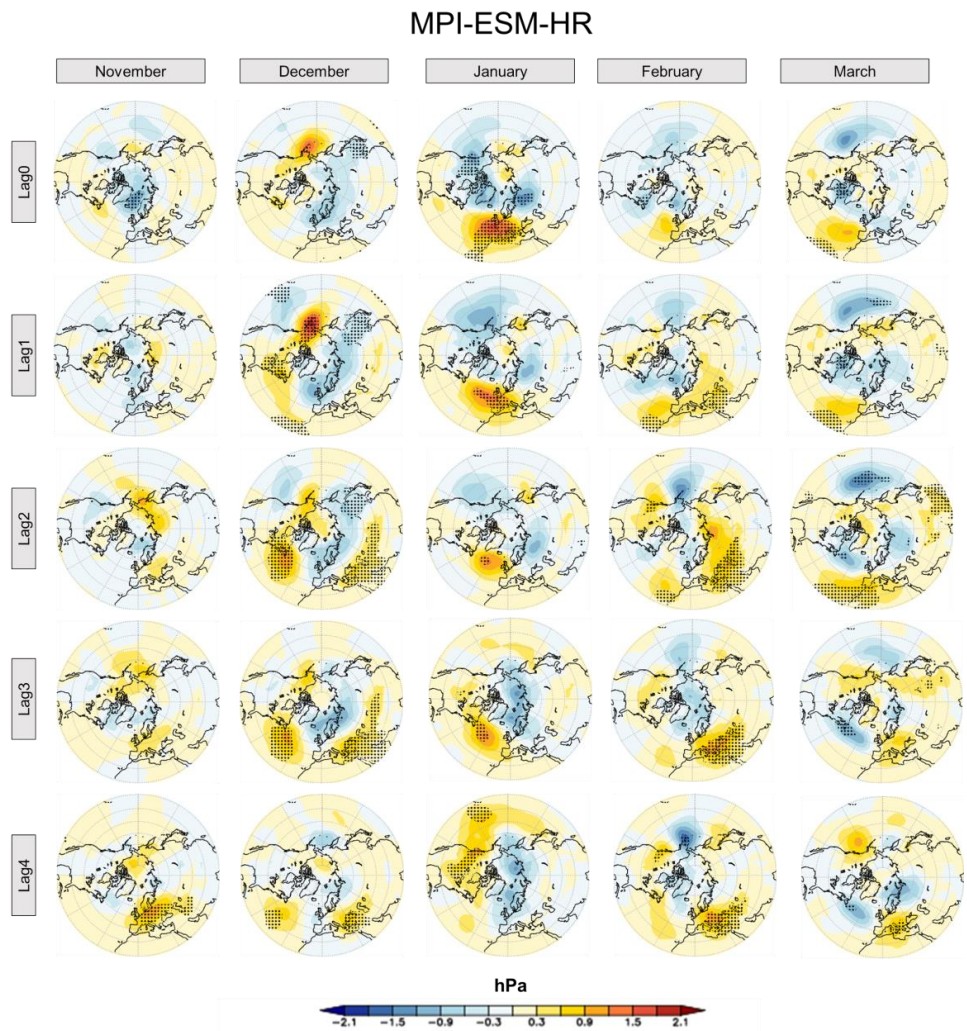


**Figure 7:** As Figure 6, but for the ensemble mean of the MPI-ESM-HR MiKlip historical simulations.


## 6 A synchronization of the NAO by the solar cycle?

In the following, we will address the question, if the quasi-decadal variations of the solar cycle have the ability to
synchronize the decadal component of the NAO, as proposed by Thiéblemont et al., (2015) and Drews et al., (2022).
For a better comparison, we apply the same analytical strategy as proposed by Thiéblemont et al. (2015) to our model
simulations and the HadSLP2 data, however with the exception that we use the SSN instead of the F10.7 solar flux





times series as a solar proxy. Since both the SSN and F10.7 time series show the same oscillations on the interannual
and decadal time scale, this is irrelevant for the interpretation of the results.  First, an EOF analysis is applied to the
deseasonalized MSLP data over the Atlantic sector (20 – 80°N, 90°W – 40°E) in the winter season (DJF averaged).
The resulting leading principal components (PC1) are then used to describe the variability of the NAO. To mute major
parts of the interannual variability, we apply a Butterworth bandpass filter with cutoff frequencies of 9 and 13 years
to the PC1 and the SSN time series. As a result, the filtered PC1 and SSN time series only include the oscillations
operating on the quasi-decadal timescale. Subsequently, lead/lag correlations are calculated between the bandpass-
filtered PC1 and SSN timeseries for both the complete dataset and all individual ensemble members (1 to 10). Drews
et al. (2022) recently argued that the correlations would become more meaningful during the course of the 20th century
due to a series of solar cycles with stronger amplitudes. We, therefore, compute the correlations for three different
time segments: the whole period (WP) (1880 – 1999), the early period (EP) with weaker solar amplitudes (1880 –
1940) and the late period (LP) with more pronounced solar amplitudes (1941 – 1999).
For the HadSLP2 dataset (Figure 8, left column/first row) positive correlations between the decadal variation of the
NAO and the solar forcing is found for the lag years one to four in both the WP and the LP periods, with maximum
correlations at lag year three during the LP. For the EP, we find an out-of-phase relation between the solar time series
and the NAO on the decadal timescale. The evaluation of this (1 ensemble member) observational dataset implies that
the solar forcing actually leads the surface response by a couple of years and that this relation is more pronounced
during phases of higher solar activity. Indeed, similar phase relations in the different time segments are given in
individual ensemble members of the MiKlip historical simulations (e.g. EM9 (Figure 8, left column/sixth row).
However, phase relations like these seem far from being a robust feature if all model runs are considered. As an
example, EM5 (Figure 8, left column/third row) indicates positive correlations between the decadal behavior of the
SSN and the NAO time series for the lag years one to three during the EP, while this relation reverses (showing
negative correlations) during the WP and LP. This is also true for EM3 (left column/third row and ) and EM7 (left
column/fifth row). Other ensemble members (EM2; Figure 8, right column/second row) suggest a maximization of the
solar impact at lag year zero and this independently of the considered period. Furthermore, EM6 (Figure 8, right
column/fourth row) indicates stronger positive correlations at positive lag years during the EP than during the LP. The
most striking discrepancies, however, come from EM1 (Figure 8, left column/second row) and EM4 (Figure 8, right
column/third row). While EM1 shows negative correlations between the solar forcing and the NAO at positive lags





(in all time segments), this is vice versa in EM4. These surface responses in EM1 and EM4 are, however, opposite to
what would be expected from the polar vortex responses in these two ensemble members (a pronounced strengthening
of the polar vortex and a downward propagation of westerly wind anomalies to the surface in EM1, and a weakening
of the polar vortex and a downward propagation of easterly wind anomalies to the surface in EM4 during winter (see
Figure 5)) and opposite to the 'top-down mechanism'.
When applied to the complete dataset of the MiKlip historical simulations, the correlation analysis yields a weak
positive (albeit significant) correlation at the lag years two to four, rather independently of the considered time
segment. This, however, should rather be interpreted as a slight (and by chance) overhang to positive correlations in
the MiKlip dataset (that could change in a larger ensemble) than a robust physical connection between the solar forcing
and the NAO. To verify whether the use of the seasonal mean (DJF) might dampen the solar cycle response, as
discussed by Drews et al. (2022), we repeated the analysis for the individual winter months (December, January and
February, see supplementary material) for the model data. We did not detect stronger connections between the decadal
solar forcing and the NAO in the calculations based on individual months compared to the seasonal mean. On the
contrary, the correlation analysis based on the December months (i.e., the month where we find the "strongest" "top-
down" signals in the middle atmosphere) depicts negative correlations at positive lag years. In summary, given all of
these inconsistencies we suspect that there is no robust connection between the quasi-decadal solar oscillations and
the respective phase of the NAO in the CMIP5 MiKlip historical ensemble simulations.








**Figure 8:** Lead/lag-correlations between the bandpass filtered PC1 based on NAO and SSN time series. For the
HadSLP2 dataset and the ensemble mean over the MPI-ESM-HR historical simulations (top row) and the individual
MPI-ESM-HR historical runs (rows 2 to 6). Green dots mark statistically significant (95%) correlations.


**7. Summary and discussion**
Our analysis of the MiKlip historical ensemble simulations, conducted with the state-of-the-art Earth system model
MPI-ESM-HR, revealed robust (and statistically significant) solar signals in the TST (see Figures 1 and 2). The
dynamical response to the initial solar temperature signal at the tropical stratopause, in the NH middle to polar latitudes
during the boreal winter season, however, showed a large spread among our data. This applies to the variability of the
PNJ and the 10 hPa zonal-mean zonal wind time series, which both did not show meaningful correlations with the
solar forcing (see Figure 3). When removing other than decadal variability components by MLR analysis, we were
able to detect (albeit rather weak) solar signals in the NH winter, in both the ensemble mean zonal-mean temperature
and zonal-mean zonal wind, that basically agree with the proposed "top-down" influence of solar variability in the
middle atmosphere (see Figure 4). However, the MLR analysis based on individual ensemble members revealed
signals of opposite direction (i.e. a strengthening (EM1) or weakening (EM4) of the polar vortex during periods of
high solar activity) (see Figure 5). Furthermore, we find indications that the detected anomalies in the zonal-mean
zonal wind at the surface are most likely independent of the signals in the middle atmosphere. The alleged surface
solar signals in MSLP seem to mimic AO-positive (and AO-negative) patterns rather randomly than in a systematic
way. This applies to the HadSLP2 data (Figure 6) and to the model data (Figure 7), which both depict most pronounced
an AO-positive pattern in January and February at different lag years and thus in months, where the strong stratospheric
influence (in December) is already weak or even reverses sign (compare Figure 4). With respect to the suggested
synchronization between the decadal solar forcing and the NAO (e.g. Thiéblemont et al., 2015) we can not find any
meaningful relations in the MiKlip historical simulations. This is supported by the fact that all ensemble members
show very individual phase relations (i.e. positive/negative correlations and maximizations during different lag years)
between the solar and the NAO time series. Additionally, more robust correlations could not be achieved in different
time segments (i.e. periods with stronger or weaker solar forcing). These findings apply to the seasonal winter mean
(DJF) as well as to individual winter months (December, January and February). As a consequence, the detected phase
relations in the HadSLP2 dataset should be interpreted carefully with respect to potential physical connections between





the solar forcing and the NAO, in particular since the observations represent only one single ensemble member.
In summary, we draw four major conclusions:

1.  The decadal variations of the TST in the MiKlip historical simulations are a product of the 11-year solar

cycle. In the course of this, an increase in the solar intensity leads to enhanced radiative shortwave heating

rates and a warming of the TST. These findings are consistent with other modeling studies concerning the

imprints of the 11-year solar cycle in the tropical upper stratosphere (Matthes et al., 2004, 2006; Schmidt et

al., 2010; Ineson et al., 2011; Chiodo et al., 2012; Langematz et al., 2013). The solar signals in the TST are

statistically significant and robust. They were detected by our correlation and MLR analyses.

2.  The dynamical response of the NH during winter in the middle atmosphere shows a weak strengthening of

the polar vortex during solar maximum in the ensemble mean in the MLR analysis. However, the signals

(especially in the zonal-mean zonal wind) are mostly insignificant and of opposite sign in individual ensemble

members, and thus not a robust feature. We suppose that the dynamical background state in the middle

atmosphere (i.e. the variability of the polar vortex) seems to play an important role for the transfer of the

initial radiative solar signal from the upper tropical stratosphere down to the troposphere in NH winter.

3.  The detected anomalies in the zonal-mean zonal wind and MSLP at the surface seem not to be related to the

seasonal march of the signals in the middle atmosphere and are most likely a manifestation of the internal

variability in the troposphere itself.

4.  Concerning the decadal variations of the NAO and the solar forcing, our results suggest that both time series

are independent from each other. In this context we find manifold phase relations throughout all of our

ensemble members, which implies a statistical by chance relation but not a physical sound connection.


Since the critical study of Chiodo et al. (2019), the "top-down" mechanism and its surface imprints have been
intensively discussed in the scientific community. It is unquestionable that early studies with GCMs and CCMs found
evidence of a "top-down" mechanism in the middle atmosphere which in most cases penetrated into the troposphere
in NH winter (Matthes et al., 2004, 2006; Schmidt et al., 2010; Ineson et al., 2011; Chiodo et al., 2012; Langematz et
al., 2013). These studies all reproduced more or less the basic features of the "top-down" mechanism, thus confirming
the physical mechanisms at work, as suggested by Kodera and Kodera (2002). In contrast, more recent simulations
with CCMs and ESMs do not seem to find statistical responses of surface variables to the decadal solar forcing (e.g.





Chiodo et al., 2019; this study). Only Drews et al. (2022) showed a near-surface solar imprint for solar cycles with
strong amplitudes. We suggest that the gradual 'fading away' of significant solar near-surface signatures in more up-
to-date model studies is closely related to progresses made in model development and computer capacities allowing
for ensemble simulations. The early simulations were conducted with fixed lower boundary conditions (i.e. prescribed
SSTs from observations or control run experiments) (Matthes et al., 2006; Marsh et al., 2007; Schmidt et al., 2010;
Chiodo et al., 2012). Some applied perpetual conditions for the solar forcing (i.e. perpetual solar maximum vs.
perpetual solar minimum) and steady-state conditions for the greenhouse gas forcing (Matthes et al., 2006; Marsh et
al., 2007; Schmidt et al., 2010; Ineson et al., 2011). In both cases, the complex nature and spectrum of internal
variability is damped. Prescribed SSTs, for example, prevent the model from developing the complete spectrum of
interannual variability in the troposphere (e.g. induced by the internal variability of the NAO), which might counteract
potential surface solar signals. In addition, steady-state background conditions in atmospheric greenhouse gas
concentrations and prescribed ozone depleting substances do not take into account transient adjustment processes in
the atmospheric dynamics, which again lead to a reduction of the overall internal variability and maybe an
overestimation of solar-induced signals. Moreover, due to more limited computer capacities, the results from the early
model studies were mostly based on single simulations.
In contrast, our results show that in a state-of-the-art climate model system the potential mean solar near-surface
signals are rather weak, not robust and inconsistent with the timing in the middle atmosphere. One potential reason is
the additional variability component introduced into the model by the interactively coupled ocean model. Misios and
Schmidt (2012) also showed the impact of an interactive ocean on the simulated solar response in the tropical Pacific
region. While individual ensemble simulations produce the expected phase correlation between the NAO and the solar
cycle, others show the opposite behavior. We, thus, do not find any convincing evidence in our model simulations of
the alleged decadal synchronization between the NAO and the solar forcing, as suggested by Thiéblemont et al. (2015).
In our view, the decadal near-surface signals detected in the MiKlip historical simulations are a product of the internal
variability in the troposphere itself and not a physical consequence of the "top-down" mechanism.
We would further like to mention that a strong reduction of the interannual variability in two basically independent
time series – be it by bandpass filtering like in our study or in Thiéblemont et al. (2015), or by using wide running
mean windows like in Drews et al. (2022) – will always lead to significant alignments of these two time series at some
point, if they are shifted towards each other gradually. Thus, the phase relations in our (and other studies) seem to be



a statistical artifact and not the consequence of a physical phase coupling. We also would like to question if the oceanic
memory is sensitive enough to store the tiny surface solar signals (even if there are some) for the duration of a complete
decade. Also, please keep in mind the strong variability of the main pressure systems in the North Atlantic, which
might wipe out potential surface solar signals within a couple of months. In our opinion, a physically sound explanation
for the alleged NAO-solar cycle phase coupling is missing so far. Thus, the claim that an inclusion of the 11-year solar
cycle would lead to a better understanding of the decadal oscillations in the NH troposphere during winter, is not
supported by our analyses of the MiKlip historical ensemble simulations. Future studies with a distinct focus on the
decadal prediction skill might help to confirm our results.

**Data availability**
The main numerical results will be made available upon request by the authors.
**Author contributions**
TS was in charge in conducting the analysis and writing the manuscript. UL initiated the study and contributed to

writing the manuscript. HP and JK were involved in conducting the MiKlip historical simulations and writing the

manuscript

**Competing interests**
The authors do not declare any competing interests.
**Acknowledgements**
We like to thank the DKRZ for granting the computational resources during MiKlip.
**Financial support**
BMBF funded projects MiKlip-2 (Förderkennzeichen 01LP1517A and 01LP1519A) and SOLCHECK
(Förderkennzeichen 01LG1906C)
**Review statement**
About to come



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
