# Peer review of "A critical evaluation of decadal solar cycle imprints in the MiKlip historical ensemble simulations"

_Weather and Climate Dynamics, 2023_

## Referee Comment (RC1)

This manuscript presents a comprehensive evaluation of the 11-year solar cycle imprints from the middle atmosphere to the surface, especially the so-called "top-down" mechanism, based on 10-member CMIP5 historical simulations from the MiKlip (MPI-ESM-HR). Considering the diverse and debated conclusion on this topic in the existing studies, I think this paper is worth to be published after a minor review. The following are my comments on this manuscript:

**Lines 17-18:** "……on the North Atlantic Oscillation might lead to ……", I would suggest writing in a way like: "……on the sea level pressure in the North Atlantic might lead to a synchronization of the North Atlantic Oscillation via the 11-year solar cycle"

**Lines 32-33:** "……with opposite sign in response……" This also suggests the large internal variability could veil the possible solar imprints.

**Lines 93-104:** all these studies are based on the same climate model, i.e. CESM-WACCM, of course, different versions, and different forcings. So even using the same climate model, the detected solar imprints or the interpreted results also could be controversial.

**Lines 134-135:** I guess the sea ice module is included in the MPIOM, right? Please add a few words here.

**Line 142**: "We focus on the period 1880-1999" Why did not use all model years (1850-2005)?

**Line 160:** How did you perform the EOF analysis on the ensemble members and the ensemble mean? Did EOF on stitching data (i.e. 1200 model years) or did EOF on each member separately, but keeping the phases (sign in EOF1) the same as the observation? I ask this question because the EOFs could be opposite signs, if performing the EOF separately, need to double-check to make sure the patterns of all the members are in the same phase.

**Section 2:** Significance level is shown in the lead-lag correlation and MLR analysis results, but the method description is missing. Considering after a band-pass filter, the effective degree of freedom of the time series will reduce a lot, so it's necessary carefully calculate the EDF and describe the method in this section.

By the way, how did you assess the robustness of the solar imprints?

**Line 174:** "180 SSN", here hard to understand this term for a reader not familiar with the MLR, please explain more.

**Line 176:** "the solar forcing lags the model data", lags? for investigating the lagged response in climate, should solar forcing **lead** the model data, right? Did I misunderstand something?

**Lines 175-176**: Are the other predictors keeping the same as the lag 0 when shifting the solar time series?

**Line 207**: "period from 1850-1999" the period from 1850-2005?

**Line 209:** "pretty high minimum" do you mean compared to the SC19? I think the response of SWHR to the minimum of SC20 fits quite well, but a higher response to the maximum of SC20. So, please clarify your statement in this sentence. )

**Figure 1:** a) and b) are missing in the figure. And I would expect a similar figure as Figure 1, but for temperature in the tropical mean (supplementary figure or panel into Figure 1), this figure will nicely show the sensitivity of the temperature response to the solar cycle amplitude.

I also suggest plotting the original SSN time series and the scaled SSN time series used in MLR (180 SSN?), as a supplementary figure.

**Line 219:** "a robust" How did you define this?

**Line 223**: "a relative short time series of satellite…" maybe provide a specific time period? I think this period probably includes different solar cycles from the simulated time series. Repeating the model analysis in the same data period as the satellite would help to compare

**Figure 2:** A very brief caption. Please add more details. Like, is it a result of MLR? or a composite of annual mean zonal mean temperature in the solar maximum years? If it's the latter, how did you define the solar maximum years (maximum of SSN of each solar cycle I guess)? What does it look like in the solar minimum?

It seems the second warming is absent in your simulations.

**Section 4**: a very long paragraph, that needs a break somewhere.

**Line 256**: What does it imply, if the SSN is below the SC14 maximum? How strong the solar cycle could be if it's above the SC maximum? I think a figure of the original SSN time series will help to give the reader directly the impression.

**Line 266:** please add a reference for this statement.

**Line 270:** As the "dynamical response of the PNJ" is directly related to the meridional temperature gradient, I'm curious, can the solar-induced TST warming really increase the poleward temperature gradient? I think it's not the case for some ensemble members, at least for the EM4.

**Figure 4:** same as Figure 2, the caption is too brief. Is it a result of MLR? MLR performed on ensemble mean or averaged regression coefficients of all the members?

Horizontal components of the E-P flux are hard to see in the bottom row of Figure 4, is it due to a very **less reflection? Maybe scaling it to show more clearly.**

**Line 339:** "insulation" Do you mean "insolation"?

**Lines 370-371**: I think the temperature response in EM4 is very different from EM1, it's almost the opposite state in the Pole region.

About Figure 5, no significant temperature response in the tropical stratopause and no response of the lower mesospheric subtropical jet in EM4. Is the warming migration to the high latitude in EM1 due to solar insolation? If yes, why it disappears in EM4?

**Line 384:** "Drews et al., (2022)" => "Drews et al. (2022)", they used a different climate model (CESM-WACCM)

**Lines 413-415 and Figure 6**: did a "top-down" mechanism show in observational data? An anomalous zonal mean zonal wind in the troposphere and surface in Feb?

**Line 444-446:** I guess that's why we need some positive (slow) feedback from the ocean. But I'm not sure your model can simulate this. By the way, if the "top-down" signals are not robust, how can we expect the surface response?

**Figure 7:** I suggest adding the surface wind or surface zonal wind in Figure 7. Same comments on the caption as above.

The spatial pattern of solar imprints in SLP may have different active centers from the EOF1 (NAO). Could you please compare them? Is the EOF1 (NAO) in your model the same as the observation?

**Lines 458-463**: this should be in the method section.

---best,

Wenjuan

---

## Author Comment (AC1)

**Reply to reviewer 1:**

Dear Wenjuan,

we would like to thank you for taking the time to carefully review our manuscript. Below you will find our response to your comments in blue indicating the respective changes/improvements where necessary.

This manuscript presents a comprehensive evaluation of the 11-year solar cycle imprints from the middle atmosphere to the surface, especially the so-called "top-down" mechanism, based on 10-member CMIP5 historical simulations from the MiKlip (MPI-ESM-HR). Considering the diverse and debated conclusion on this topic in the existing studies, I think this paper is worth to be published after a minor review. The following are my comments on this manuscript:

Lines 17-18: "……on the North Atlantic Oscillation might lead to ……", I would suggest writing in a way like: "……on the sea level pressure in the North Atlantic might lead to a synchronization of the North Atlantic Oscillation via the 11-year solar cycle"

**Rephrased:** "More recently it has been proposed that the projection of possible top-down initiated decadal solar signals on surface climate might modulate and synchronize the North Atlantic Oscillation."

Lines 32-33: "……with opposite sign in response……" This also suggests the large internal variability could veil the possible solar imprints.

*Indeed! We stress the potential impact of the internal variability in different atmospheric domains and ensemble members with respect to possible solar signals throughout the manuscript.*

**Added:** "which might be a result of the large overall internal variability compensating rather small solar imprints."

Lines 93-104: all these studies are based on the same climate model, i.e., CESM-WACCM, of course, different versions, and different forcings. So even using the same climate model, the detected solar imprints or the interpreted results also could be controversial.

*Thank you for pointing this out, since it makes our interpretation even stronger!*

**Added:** "In the context of the most recent literature, it is difficult to understand why Chiodo et al. (2019) and Drews et al. (2022) arrive at a different assessment of the solar signal, even though the same model was used. This might point to the fact, that the complexity of the model is not the most relevant component in shaping potential surface solar signals, but rather the effects of internal variability in individual model runs and (to some degree) the applied analysis"

Lines 134-135: I guess the sea ice module is included in the MPIOM, right? Please add a few words here.

*MPIOM includes sea ice dynamics, like any complex ocean model. The physical description of the sea ice model can be found in the cited literature. We think a more detailed description is not necessary or expedient.*

Line 142: "We focus on the period 1880-1999" Why did not use all model years (1850-2005)?

*We chose to analyze the 1880-1999 period, since the very early Hadley observations (both SSTs and MSLP) are only little reliable. Additionally, the MPI-ESM model was spun up using a constant solar forcing, which might influence the oceanic response prior to 1880.*

Line 160: How did you perform the EOF analysis on the ensemble members and the ensemble mean? Did EOF on stitching data (i.e. 1200 model years) or did EOF on each member separately, but keeping the phases (sign in EOF1) the same as the observation? I ask this question because the EOFs could be opposite signs, if performing the EOF separately, need to double-check to make sure the patterns of all the members are in the same phase.

*Thank you for pointing this out! We were aware of this problem and thus we stitched the input data and then performed the EOF analysis. This prevents our analysis from sporadic sign conversions.*

Section 2:Significance level is shown in the lead-lag correlation and MLR analysis results, but the method description is missing. Considering after a band-pass filter, the effective degree of freedom of the time series will reduce a lot, so it's necessary carefully calculate the EDF and describe the method in this section.

*Correct! The description was not precise enough so far!*

**Added:** "All correlation analyses have been performed by using the Python scipy.pearsonr function. Statistical significance of the correlations has been estimated by using a two-tailed Student's t-test, as implemented in Python."

By the way, how did you assess the robustness of the solar imprints?

*We use the term robust in a non-statistical sense. We declare a result to be robust if, for instance, a comparable reaction in a meteorological variable (e.g., the stratopause temperature) is present in the majority of our ensemble members.*

Line 174: "180 SSN", here hard to understand this term for a reader not familiar with the MLR, please explain more.

*We agree.*

**Added: "**Based on this MLR analysis, we derived the model response to our chosen set of predictors, e.g., the temperature response per unit of the predictor (i.e., K per 1 SSN).  To display the model response during solar maximum we scaled the coefficients to 180 SSN, which is a good approximation for a mean solar cycle amplitude between 1880 and 1999."

Line 176: "the solar forcing lags the model data", lags? for investigating the lagged response in climate, should solar forcing lead the model data, right? Did I misunderstand something?

*It depends on the interpretation of leads and lags. As an example, if we would like to investigate the response of a meteorological variable one year after the solar maximum, we would analyze the model response in the year 1990 as a result of the solar forcing in the year 1989. Thus, the solar timeseries lags the model data by 1 year in this example. However, it is true that you could also say that we compare the model response to the solar forcing one year after the solar maximum. In this case we would analyze the lagged model response to the solar forcing.*

*The description, suggested by the reviewer is maybe more intuitive thus we changed the sentence.*

*Rephrased: "… the solar time series has been shifted in such a way that the model response lags the solar forcing by 1 to 4 years."*

Lines 175-176: Are the other predictors keeping the same as the lag 0 when shifting the solar time series?

*Yes. Otherwise, we would mix up signals.*

Line 207: "period from 1850-1999" the period from 1850-2005?

*Corrected. Replaced by "over the period 1880 – 1999"*

Line 209: "pretty high minimum" do you mean compared to the SC19? I think the response of SWHR to the minimum of SC20 fits quite well, but a higher response to the maximum of SC20. So, please clarify your statement in this sentence. )

*Thank you for pointing this out. The original sentence had been misplaced. It has been replaced by:* "Only during SC20, the maximum SWHR response is higher than expected for that weak solar cycle."

Figure 1: a) and b) are missing in the figure. And I would expect a similar figure as Figure 1, but for temperature in the tropical mean (supplementary figure or panel into Figure 1), this figure will nicely show the sensitivity of the temperature response to the solar cycle amplitude. I also suggest plotting the original SSN time series and the scaled SSN time series used in MLR (180 SSN?), as a supplementary figure.

*Unfortunately, only one EM was available with a diagnostic output for the swhr. Thus, producing a similar plot like Figure 2 is not possible. Saying this, we like to point out that one EM is enough to analyze the swhr, since they show the direct radiative response (and ozone is a prescribed variable in the MiKlip simulations) and the result will be identical among the different EMs. The sensitivity of the temperature response is shown and discussed in detail based on Figure 3.*

**Added:** "a and b", and a Figure showing the original SSN time series (there is no scaled SSN time series available, since only the resulting coefficients are scaled not the SSN time series itself (please see comment above).

Line 219: "a robust" How did you define this?

*Please see above.*

Line 223: "a relative short time series of satellite…" maybe provide a specific time period? I think this period probably includes different solar cycles from the simulated time series. Repeating the model analysis in the same data period as the satellite would help to compare.

*We like to show the model response over the complete period here, since it is important for the rest of the analysis. Since we don't know for sure the reason for these slightly higher values, we removed the statement.*

Figure 2: A very brief caption. Please add more details. Like, is it a result of MLR? or a composite of annual mean zonal mean temperature in the solar maximum years? If it's the latter, how did you define the solar maximum years (maximum of SSN of each solar cycle I guess)? What does it look like in the solar minimum?

*The result is based on MLR, as said in the caption, and thus the temperature response is scaled as described above. To derive the temperature profile during solar minimum, composite analysis could be an option. This, however, is not the scope of this manuscript.*

It seems the second warming is absent in your simulations.

*The long discussed secondary peak is, indeed, not present. However, we would like to point to most recent literature where the so-called secondary peak can no longer be found even in satellite data. Dhomse et al. (2022) suggest that the secondary peak (found in earlier studies) emerged most likely due to aliasing effects related to the Mount Pinatubo eruption in 1991 and probably was not a result of solar variability.*

Section 4: a very long paragraph, that needs a break somewhere.

*We created a new paragraph that starts with:* "After having analyzed the variability of the TST, the PNJ and the 10 hPa zonal-mean zonal wind, we will now isolate potential solar signals by the aid of MLR."

Line 256: What does it imply, if the SSN is below the SC14 maximum? How strong the solar cycle could be if it's above the SC maximum? I think a figure of the original SSN time series will help to give the reader directly the impression.

*Included. Please see comment above.*

Line 266: please add a reference for this statement.

**Added:** "(Butchart, 2014).

Line 270: As the "dynamical response of the PNJ" is directly related to the meridional temperature gradient, I'm curious, can the solar-induced TST warming really increase the poleward temperature gradient? I think it's not the case for some ensemble members, at least for the EM4.

*In our understanding the individual internal variability of each EM will determine if the (relatively weak) solar signals can shine through in the middle atmosphere or not. To clarify if this is the case, one would have to analyze the internal conditions of each individual ensemble member with respect to the polar vortex dynamics using very comprehensive analysis. In our paper, we would like to open the discussion and hope that future analysis will provide more insights on this.*

Figure 4: same as Figure 2, the caption is too brief. Is it a result of MLR? MLR performed on ensemble mean or averaged regression coefficients of all the members?

*Done.*

Horizontal components of the E-P flux are hard to see in the bottom row of Figure 4, is it due to a very less reflection? Maybe scaling it to show more clearly.

*The EP-flux vectors have already been scaled for a better visualization. The horizontal component is just much smaller than the vertical component. Scaling the horizontal component would lead to artificial results. Thus, we would like to refrain from doing this.*

Line 339: "insulation" Do you mean "insolation"?

*Corrected.*

Lines 370-371: I think the temperature response in EM4 is very different from EM1, it's almost the opposite state in the Pole region.

*Agreed and this, again, points to the importance of the internal variability in each model run.*

About Figure 5, no significant temperature response in the tropical stratopause and no response of the lower mesospheric subtropical jet in EM4. Is the warming migration to the high latitude in EM1 due to solar insolation? If yes, why it disappears in EM4?

*Agreed. If this signal would only be related to the march of the solar insolation, it should be present in all EMs. It is, however, possible that this signal is overprinted by the internal variability in this individual model run.*

Line 384: "Drews et al., (2022)" => "Drews et al. (2022)", they used a different climate model (CESM-WACCM)

*Corrected. Yes, they used a different model but the statements in this publication were rather general in our opinion. And as pointed out by the reviewer above, even though using the same model it did not lead to comparable results.*

Lines 413-415 and Figure 6: did a "top-down" mechanism show in observational data? An anomalous zonal mean zonal wind in the troposphere and surface in Feb?

*It is not possible to answer this question since the results, depicted in Figure 6, include the analysis for the period 1880-1999 and middle atmosphere observations are not available over this complete period. This, however, is not only an obstacle in our manuscript but also in previous papers (e.g., Gray et al., 2013). Please also keep in mind that the observations only represent one 'ensemble member' which can lead to a misinterpretation (or overinterpretation) of potential solar signals.*

Line 444-446: I guess that's why we need some positive (slow) feedback from the ocean. But I'm not sure your model can simulate this.

*We use one of the most widely used and complex general ocean circulation models, the MPIOM. So, we are rather confident about its general performance. Additionally, we don't think that the ocean component is responsible but the fact that the solar signals are very small and (if at all) occur very irregularly, which can not lead to a regular phase relation. The ocean is just not sensitive enough to preserve these tiny signals over a longer period of time. Even very strong signals (e.g., anthropogenic or volcanic signals) are hard to detect in ocean dynamics. Furthermore, a detailed description of the underlying physics, forming the basis for the alleged lagged responses, is missing in literature.*

By the way, if the "top-down" signals are not robust, how can we expect the surface response?

*We can't and this one of the main points of this paper. If the "solar signals" are already extremely small, irregular and hard to detect in the domain of the middle atmosphere (maybe due to the high internal variability there) the surface signals detected by MLR, composites or any other technique only capture the temporal state of the troposphere itself (sometimes NAO(+), sometimes NAO(-)) independent of the solar variability. In our opinion, future work should focus more on the inconsistencies in the middle atmosphere than directly discuss possible small surface signals or even a synchronization of hemispherical scale climate modes via the solar cycle. These points are given in the discussion section.*

Figure 7: I suggest adding the surface wind or surface zonal wind in Figure 7. Same comments on the caption as above.

*As the "problems" already start to appear in the middle atmosphere, including more tropospheric variables is not expedient in our opinion. The MSLP is pretty much the standard metric when discussing surface solar signals.*

The spatial pattern of solar imprints in SLP may have different active centers from the EOF1 (NAO). Could you please compare them? Is the EOF1 (NAO) in your model the same as the observation?

*Yes, we compared the spatial EOF1 pattern of the model and the observations. We include a figure below. The centers of action are comparable in both the model and the HadSLP2 data. In general, the pattern is more dominant in the model, however, please keep in mind that the observations only include 120 model years, while the model runs consist of 1200 years in total. A less dominant pattern might also be found in individual EMs.*

Lines 458-463: this should be in the method section.

*It is already included in the method section. We just remind the reader again here before discussing the results.*

[Figure]

**Figure for reviewer 1**: EOF1 pattern (DJF) for HadSLP2 and MPI-ESM-HR data. Both calculated over the period 1880-1999

**Literature**

Dhomse, S. S., Chipperfield, M. P., Feng, W., Hossaini, R., Mann, G. W., Santee, M. L., & Weber, M. (2022). A single-peak-structured solar cycle signal in stratospheric ozone based on Microwave Limb Sounder observations and model simulations. *Atmospheric Chemistry and Physics*, *22*(2), 903-916.

Gray, L. J., Scaife, A. A., Mitchell, D. M., Osprey, S., Ineson, S., Hardiman, S., ... & Kodera, K. (2013). A lagged response to the 11 year solar cycle in observed winter Atlantic/European weather patterns. *Journal of Geophysical Research: Atmospheres*, *118*(24), 13-405.

---

## Author Comment (AC2)

**Reply to reviewer 2:**

Dear Reviewer,

we would like to thank you for taking the time to carefully review our manuscript. Below you will find our response to your comments in blue indicating the respective changes/improvements where necessary.

Spiegl et al. examine the impact of the 11-year solar cycle on decadal predictions, by using a large data-set from the MPI decadal prediction system. The length and size of their ensemble experiments offer an unprecedented opportunity to critically evaluate the possibility (suggested in many other studies) concerning a possible role of the solar cycle as a source of predictability (via the "top-down" mechanism). This is a very controversial and yet relevant topic, given the potential implicatuions for decadal prediction of boreal wintertime climate in the North Atlantic. This paper nicely shows that the top-down signal in the MPI model is small and there is hardly any synchronization of the NAO. In particular, the paper also debunks the possibility raised in another recent paper (Drews et al., 2022) that the time-dependence of the solar/NAO signal highlighted in Chiodo et al. (2019) might only arise in decades with a stronger solar cycle. The paper nicely shows that indeed, the modeled response is weak and mostly not significant and indeed, the main conclusion against a solar modulation of the NAO and synchronization for their model is absolutely warranted. I think that the paper will contribute to the scientific debate on this matter and deserves to be published, after some corrections as detailed below.

GENERAL COMMENTS

- The main conclusions of the paper are supported by the evidence provided in the paper. However, it would be good to at least acnowledge and/or discuss the possibility that the MPI-ESM-HR model system, just like other model systems, might suffer from the "signal-to-noise" paradox issue, according to which, the models might underestimate any externally forced signals in dynamical aspects of the climate system (see e.g., Scaife and Smith, https://www.nature.com/articles/s41612-018-0038-4), such as the PNJ and stratosphere-troposphere coupling.

Thanks for this important point! Added the following paragraph to the discussion section:

**Added**: "The discrepancies between the observed and modelled internal variability in response to external forcings (such as solar variability) may also be attributed to the "signal-to-noise" paradox which states that relatively small changes in the external forcing will not lead to detectable changes in the variability spectrum in both the real climate system and model simulations as discussed by Scaife and Smith (2018)."

- It might be also good to at least discuss the implications of the part of the solar forcing which is missing in these runs; the particles forcing (MEE/SEP). I would argue that particles are unlikely to change the picture and the EPP signal is perhaps even more complicated than the irradiance/UV signal, and the models are at their infancy in simulating this forcing... but it needs to be at least discussed, given the number of related publications arguing that EPP play an important role in the solar influences on the polar vortex/ NAM and NAO.

*The effects of solar energetic particles (SEP) and medium energy electrons (MEE) were not explicitly included in the model, as in our opinion – as also mentioned by the reviewer -  these effects are unlikely to change our results, since they probably are (if present) even smaller and more complex than the alleged 11-year solar cycle surface signals. On the other hand, the model uses the observed ozone time series as a boundary condition, which might have been influenced by SEP and MEE.*

*To complete the discussion we added:*

***Added:"*** While the model simulations include both, changes in the total solar irradiance (TSI) and spectral solar irradiance (SSI), potential effects related to solar energetic particles (SEP) and medium energy electrons (MEE) are not explicitly included in the MiKlip experiments. Observations and model studies suggest that changes in the stratospheric composition related to SEP can lead to a radiatively driven modulation of the middle atmosphere dynamics, which can penetrate to lower atmospheric layers down to the troposphere (e.g., Seppälä et al., 2009, 2014; Baumgaertner et al., 2010; Arsenovic et al., 2016). However, since no robust surface impacts have been simulated even for strong solar energetic particle events (SEP) of the recent decades (Jackman et al., 2009), we infer that including these effects may not alter our results significantly.

The other general point concerns the lengthy discussions of the individual months. I think many of the parts of sections 5-6 could be shortened, especially if there's no robust signal to document.

*We prefer to not shorten Sections 5 and 6 for the sake of completeness and to avoid misinterpretation given the complexity of the subject and the ongoing discussions in the community.*

-Lastly, the paper examines the top-down mechanism in detail and rules out a robust effect in their model... but what about any bottom-up impacts? They should be at least mentioned somewhere in the discussion, to put this paper in the broader context. The existing literature is cited but there could be more discussion of the comparison with Chiodo et al., 2019 and the aspects that are new here (i.e. testing the link between the time-dependence of the signal and the magnitude of the solar cycle amplitude).

*Indeed, the paper mainly discusses possible top-down induced solar signals. However, since the model also includes TSI variations (the main driver for eventual bottom-up mechanisms), bottom-up effects would be included if they shape the decadal "surface solar signal". However, we can't find any systematic pattern in the surface meteorological variables nor the NAO. Additionally, to our knowledge, there is no literature available that addresses potential bottom-up effects in the North Atlantic Sector, since profound TSI effects are mainly limited to the low latitudes. We now mention bottom-up effects in the discussion section and use the argumentation as above. Additionally, we explicitly compare the strategy of our study to Chiodo et al., (2019) and include the point mentioned by the reviewer above.*

**Added**:

" The MiKlip simulations are more in line with Chiodo et al. (2019), who argued that the alleged surface solar signals could be an incidental product which is only detectable during phases with stronger solar cycles. Our results even suggest that robust solar surface imprints are basically absent throughout the complete historical period and are thus not sensitive to the amplitude of individual solar cycles."

*and*

*"It should be noted, that we did not explicitly analyse a potential TSI controlled bottom-up effect on the solar surface signal, as bottom-up effects are rather confined to tropical latitudes with a prolonged influence of the TSI throughout the year (Meehl et al., 2008)."*

SPECIFIC COMMENTS

L17-19: could add that previous work didn't really do a decadal prediction set-up --> rather, continuous long-term climate runs!

*We agree. A short note has been added to the abstract.*

L20 - it's more than just "confirmation" from other modeling groups - a cleaner model study is really missing...!

*Adjusted.*

L22 we aim for an unbiased evaluation --> "unbiased" seems a bit too harsh and indirect criticism towards previous studies --> replace with "objective and improved"...?

*Done.*

L30 is rather weak -> suggest removing "rather"

*Done.*

L31 remove "basically"

*Done.*

L36-37 unclear what the driver of the "anomalies" is - a bit more clarity would be needed. Would these be ensemble mean anomalies (or anomalies in the individual ensembles) that are correlated with the solar cycle, or what is the "driver" of these anomalies..?

*Rephrased this paragraph for more clarity.*

**Rephrased**:" *We find that the westerly wind anomalies in the lower troposphere as well as the anomalies in the mean sea level pressure are most likely independent from the timing of the seasonal march in the middle atmosphere and thus alleged top-down influences. The pattern rather reflects the decadal internal variability of the troposphere, mimicking positive and negative phases of the Arctic- and North Atlantic Oscillation throughout the year sporadically, which are then assigned to the solar predictor time series without any physical plausible connection and sound solar contribution."*

L36 "most likely independent from the seasonal march in the middle atmosphere" -> would this mean that the timing of the tropospheric anomalies does not match the downward propagation of the (apparent) solar signals? If so, I'd reword this to use the word "timing" so that it's clearer to the reader what we are talking about.

*See above.*

L38 "rather sporadically than in a systematic way" -> sporadic in that they depend on the ensemble member, or sporadic in that they depend on the time window being analysed? again, more specificity might be appreciated.

*See above.*

L40 "might rather be interpreted" - I'd repharase to "might be". Just remove "interpreted" to simplify the wording of this part of the abstract.

*Done.*

L41 "between the solar forcing" - it might be good to again be more specific and say that the paper handles the UV/irradiance forcing specifically... as there might be also other pathways for a solar influence on climate, namely via MEE/particle forcing. Just add "irradiance" or "UV" to "solar forcing"

*Done.*

L40 "as a statistical artefact" - it may well be that the synchronization is simply due to internal dynamics of the atmosphere...? if that's the case, then it may not be a "statistical artifact" in the strict sense, but rather, that it's not solar driven...

*Yes, that's what we mean.*

*The wording has changed to: "statistical artefact, affected for example by the internal decadal variability of the ocean... "*

L73 "...convergence in the Eliassen-Palm flux (EPF)...positive wind anomaly" a convergence in EPF would result in a deceleration of U. I guess what is meant here is thus the "divergence" of the EPF?

*Yes, correct. Thank you for spotting this.*

L77 - Discussion of Matthes 2004 - I'd be more specific here and say that this study specifically studied the evolution of the signal on sub-monthly time-scales

*Matthes et al. (2004) analyzed monthly means, while Matthes et al. (2006) used monthly subperiods to derive the downward transfer of the solar signal into the troposphere. A short note has been added.*

L81 "from February on" -> from February onwards

*Done.*

L83 "very individual temporal progressions" - again, I'd use the word "timing" to clarify what is meant here. E.g. you could reword this to "the exact timing of the downward propagation depends on the individual study"

*The sentence has been changed as suggested.*

Side note: I'm not sure Marsh '07 really show a downward solar impact - rather, they show, by means of a time-slice experiment, that the signal is robust in the upper stratosphere but in the polar regions, their signal is rather weak and non-significant...

*The reviewer is correct! What Marsh et al. (2007) are showing is the long-term annual mean response to the solar cycle. Based on this, no conclusions can be drawn with respect to the "top-down mechanism". Thus, this citation has been removed.*

around L90: it might be good to add somewhere here that there are also other potential pathways for a solar influence on climate but the irradiance-UV component is the most studied one and the one that models have an easier time simulating...?

*We agree that there are other pathways of the solar signal. This paper, however, is about the top-down mechanism and the North Atlantic sector. Discussing e.g., bottom-up effects which "might" be relevant in tropics seem misplaced in this context.*

L110 I think you could also add here that the availability of the large ensembles with the observed time-varying solar cycles allows you to explicitly test the dependence of the signal on the solar cycle amplitude (hypothesis put forward by Drews et al 2022).

*We have added this, thanks.*

L209 I'd suggest removing unscientific terms such as "pretty".

*The sentence has been removed and replaced.*

L219 replace "receive" with "obtain"

*Done.*

Figure 2 seems a bit blurry and the individual components of the figure (e.g. colorbar and right panel) seem to have been manually assembled together. I'd suggest including a high resolution version of this figure and use another software to produce the figure without having to manually assemble the individual panels. This also concerns the other Figs. 3-8.

*The reviewer is correct! Figure 2 has not been ideal yet! Replaced by another version. With respect to the other figures, we included the grey bars manually and we would like to keep this style. We slightly adjusted the arrangement of the subplots though. If the individual plots seem not to be high resolution figures in the manuscript yet, this would be due to the fact that the figures have been copied/paced from .eps and then converted to .pdf. The final figures that will be uploaded separately at the end will of course be in higher resolution to meet the quality standards of WCD.*

Figure 3 nice figure, but I have the impression that the data is not uniformly scattered in the x-dimension (Sunspot number) - the dots are all structured in "vertical bands". Why is that happening? Is the SSN sampled only every 5-10 units...? Also, do dots correspond to each individual member?

*There is a misunderstanding. The dots do not only correspond to individual ensemble members but represent all model years. The data are also not only sampled only every 5-10 units. The "vertical bands" appear because the same SSN time series (for each ensemble member) has been used for correlation. The response, however, is individual in each ensemble member. This can be seen best during the SC19 maximum, which is characterized by the highest SSN number throughout the historical epoch. This SSN value appears exactly once in 120 years (lower values appear more often), thus exactly 10 dots are arranged around this value (sometimes stronger, sometimes weaker responses). The x-axis would be correct though.*

L410-440 (and intro) I think it would also be good to add the most recent paper by Gray 2016 to this discussion... and in particular, Ma et al., 2021 - who argued that the early-winter signal is the most robust component of the signal.

*Since Gray et al., 2016 is primarily about blocking frequencies, we added Ma et al. (2018) and added it to the discussion of the early winter influence. We could not find a paper of Ma et al. (2021) dealing with solar variability and the North Atlantic sector.*

*Added:" A most recent study again concludes that the most pronounced solar signal seem to appear in early winter (Ma et al. (2018))."*

L441 I think you can be more confident here and replace "assume" with "conclude"

*Done.*

L444-447 What about the "bottom up" signal? I agree that much of this discussions disqualifies the top-down... but can we also rule out a role of any direct surface (TSI-driven) signals?

*Please see in the main comments above and thereafter.*

L544-546 Could the authors provide some evidence to support the interpretation that the background state determines if the signal is transferred or not?

*In our opinion (and at this stage) the very different "solar signals" in the middle atmosphere that we observe in individual ensemble members, even though they have been driven with the exact same external parameters, could already be a hint to the importance of the middle atmosphere dynamics. A more detailed investigation is currently part of the SOLCHECK project. The following sentence has been added: "The important role of middle atmosphere dynamics in modulating potential solar signals is currently investigated as part of the SOLCHECK project and will be published in a subsequent paper (Wenjuan Huo, personal communication)."*

L550 remove "time series"

*Done.*

L551 not sure what is meant with "manifold phase relations". can the authors clarify?

*We mean different NAO-solar forcing phase relations, as shown in Figure 8. The sentence has been clarified: "We find a range of phase relations between the NAO and the solar forcing throughout our ensemble members, which implies a random statistical relation rather than a physical sound connection."*

L563 "...made in model development.." agreed that having larger ensembles helps diagnosing robust signals and probably that's partly the reason why signals in older studies were deemed as "solar"... on the other hand, I do not think that model physics being simpler in those early studies is another reason for the apparently stronger signal. Those models had all the necessary physics to capture the mechanism, in principle.

*We fully agree. The early GCMs included the necessary UV-radiation codes and middle atmosphere dynamics to simulate the solar top-down signal. However, to study the tropospheric signal, more components had to be added to the models, in particular ocean models. This together with ensemble realizations nearly inhibits the identification of a robust surface solar signal.*

*We extended the sentence to: "While these models disposed of the necessary physical mechanisms, i.e., UV radiation codes and middle atmosphere dynamics, to capture the solar UV-induced top-down solar signal, the complex nature of physical and chemical processes and the spectrum of internal variability were reduced."*

L583 I'd add somewhere here that these results agree with the conclusions of Chiodo et al., 2019 - and here, they even strengthen those conclusions concerning the little role of solar forcing in modularing the NAO, as the runs of this paper can even more directly be compared with observations and the solar signal can be tests in an even more realistic set-up (with transient forcings, transient solar cycle, etc.)

*Please see our comment, related to the now included comparison to Chiodo et al. (2019) above. Additionally, we added the important point mentioned by the reviewer that the MiKlip simulations are more realistically, since they include all observed forcings over the historical period.*

---

## Author Response (AR2)

**Reply to the editor:**

Dear Thomas Birner,

we would like to thank you for taking the time to handle and carefully review our manuscript. Below you will find our response to your comments in blue indicating the respective changes/improvements where necessary.

Dear Tobias Spiegl et al,

thank you for submitting detailed responses to all reviewer comments and for preparing a revised manuscript. My assessment is that your responses and revisions mostly address the reviewer comments well and that the paper is essentially acceptable for publication. I do have a small number of minor follow-up revisions that I think would improve the presentation of your material further. Once you've addressed these further revisions I expect to be able to accept your manuscript for publication in WCD.

One particular suggestion concerns clarifications and misunderstandings by the reviewers that in some cases where only answered in the responses to reviewers without any changes to the manuscript. I suggest to treat those reviewer remarks as representative of other potential readers from the community and to reconsider adding short clarification remarks in the text. Chances are that they will help other readers.

*Thank you for this important hint! As suggested, we include more information now for more clarity (please see below).*

Specifically, I have the following reviewer comments in mind:

Reviewer 1
- usage of "robust" - certainly helpful for other readers to state how you use this descriptor
*As suggested, we added a comment to section 2.1.*

*Added:" In this manuscript we use the term "robust" if a signal of the same sign (e.g., the temperature response at the tropical stratopause) appears in the majority of our ensemble members."*

- remark on line 142 ("Why did not use all model years (1850-2005)?")
*Included now.*

*Added: "Since especially the very early years are little reliable in observations and the model has been spun-up with a constant solar forcing, we focus on the period 1880 – 1999."*

- remark on Fig. 2 ("It seems the second warming is absent in your simulations.")
*Added more discussion and Dhomse et al. (2022).*

*Added:" In our simulations we can't find the sometimes observed secondary peak in the temperature profile in the lower stratosphere. This secondary peak, however, can no longer be found even in most recent analysis of satellite data. Dhomse et al. (2022) suggest that the secondary peak (found in earlier studies) emerged most likely due to aliasing effects related to the Mount Pinatubo eruption in 1991 and probably was not a result of solar variability."*

- remark on Fig. 5 why there's no significant response in EM4

*The individual responses to the solar cycle (including statistical significance) is a result of different internal variability in the middle atmosphere in individual model runs. A more detailed paper on this will follow, where SOLCHECK simulations (including even more model years and different models) will be exploited (personal communication: Wenjuan Huo).*

*Reformulated:" However, even though exactly the same solar forcing has been applied in EM4 as in EM1, the initial temperature signal is not significant (most likely due to the individual internal variability in this ensemble member) and the dynamical response of EM4 in the extratropical regions looks very different."*

- question about "top-down" mechanism in observational data

*Included the relevant information.*

*Added:" Furthermore, and due to the lack of data covering the whole atmospheric domain over the complete historical period, it is not possible to connect the potential surface solar signals to the seasonality in the middle atmosphere. This applies to our and the original studies (e.g., Gray et al., 2013)."*

- question about EOF1 (NAO) in model (related to Fig. 7)
*Added a respective sentence.*

*Added:" Before continuing, we compared the spatial pattern of the EOF1 between the modelled and observed data and found good agreement with respect to the centers of action and overall characteristics (not shown)."*

Reviewer 2
- "vertical bands in Fig. 3"

*For more clarity we added "Please note, that the same SSN time series has been used for the correlation for all individual ensemble members, leading to a "vertical arrangement" of the data in the scatter plots shown in Figure 3."*

Some further comments:

Reviewer 1 question about effective degrees of freedom in the MLR analysis (due to band-pass etc) was not really addressed - please include relevant information in the text.

There is some misunderstanding! The data used for the MLR have not been bandpassed or comparable. We use the raw data here as in earlier papers. We only bandpass the NAO and solar timeseries to derive the decadal component before correlation and applying the ttest (as mentioned in the manuscript). This is the same strategy as in the original paper of Thieblemont et al. (2015).

*For more clarity we added "We like to note, that we use the raw (unfiltered) model output as input for our MLR analysis."*

The added text in response to Reviewer 2 remark about signal-to-noise paradox seems confusing to me. Specifically, isn't your statement "that relatively small changes in the external forcing will not lead to detectable changes in the variability spectrum in both the real climate system and model simulations" in conflict with the reviewer remark that "models might underestimate any externally forced signals in dynamical
aspects of the climate system"? Please reformulate.

*Thank you for spotting this.*

*Reformulated:" We would finally like to note that the detection of a significant decadal solar impact on the NAO in winter in the MPI-ESM-HR climate model, as in other climate models, might to some degree suffer from the 'signal-to-noise paradox', i.e., a low strength of predictable signals vs. a relatively high level of agreement between modelled and observed variability of the atmospheric circulation, which is particularly evident in the climate variability of the Atlantic sector (Scaife and Smith, 2018)."*

In some Fig. captions you state the meaning of hatching to "mark the 95% level of significance" -- please use the more precise phrase "statistical significance" to avoid misunderstanding.

*Thanks and changed accordingly.*